# Intranasal Delivery of Nanoformulations: A Potential Way of Treatment for Neurological Disorders

**DOI:** 10.3390/molecules25081929

**Published:** 2020-04-21

**Authors:** Salman Ul Islam, Adeeb Shehzad, Muhammad Bilal Ahmed, Young Sup Lee

**Affiliations:** 1School of Life Sciences, College of Natural Sciences, Kyungpook National University, Daegu 41566, Korea; dr_ssulman@yahoo.com (S.U.I.); mbilalknu@gmail.com (M.B.A.); 2Department of Clinical Pharmacy, Institute for Research and Medical Consultations (IRMC), Imam Abdulrahman bin Faisal University, Dammam 31441, Saudi Arabia; asmsiar@iau.edu.sa

**Keywords:** neurological disorders, Parkinson’s disease, Alzheimer’s disease, glioblastoma, epilepsy, multiple sclerosis, nose-to-brain, blood brain barrier, nanoformulations

## Abstract

Although the global prevalence of neurological disorders such as Parkinson’s disease, Alzheimer’s disease, glioblastoma, epilepsy, and multiple sclerosis is steadily increasing, effective delivery of drug molecules in therapeutic quantities to the central nervous system (CNS) is still lacking. The blood brain barrier (BBB) is the major obstacle for the entry of drugs into the brain, as it comprises a tight layer of endothelial cells surrounded by astrocyte foot processes that limit drugs’ entry. In recent times, intranasal drug delivery has emerged as a reliable method to bypass the BBB and treat neurological diseases. The intranasal route for drug delivery to the brain with both solution and particulate formulations has been demonstrated repeatedly in preclinical models, including in human trials. The key features determining the efficacy of drug delivery via the intranasal route include delivery to the olfactory area of the nares, a longer retention time at the nasal mucosal surface, enhanced penetration of the drugs through the nasal epithelia, and reduced drug metabolism in the nasal cavity. This review describes important neurological disorders, challenges in drug delivery to the disordered CNS, and new nasal delivery techniques designed to overcome these challenges and facilitate more efficient and targeted drug delivery. The potential for treatment possibilities with intranasal transfer of drugs will increase with the development of more effective formulations and delivery devices.

## 1. Introduction

Improving the prognoses of diseases of the central nervous system (CNS) such as Parkinson’s diseases (PD), Alzheimer’s diseases (AD), and brain tumors has always been a greater challenge than those of diseases affecting other organs [1,2,3]. Reportedly, >90% of newly proposed CNS drugs have not been approved by the US Food and Drug Administration (FDA) [4]. The presence of the complex blood brain barrier (BBB) that limits drug entry to the CNS region is the major obstacle for the development of CNS treatments [5]. Moreover, non-targeted delivery of diagnostic reagents or therapeutic drugs is known to cause significant damage to neurons and glial cells. Therefore, novel delivery platforms bearing the therapeutic drugs for neurological disorders are urgently needed.

In this era, imaging agents or treatments for CNS diseases are highly dependent on nanomedicines, because they play a promising role in CNS drug delivery. It has been shown that nanomedicines can actively and effectively cross the BBB and deeply penetrate the diseased brain tissues. In addition, nanomedicines are also associated with increased strength, stability, surface area, and sensitivity [6,7]. Novel, advanced, and versatile CNS nanomedicines can simultaneously serve diagnostic and therapeutic functions; however, a number of optimizations are still required for future widespread clinical applications.

The nasal passage is the brain’s only contact with the external environment. Upper posterior segments of the nose are connected with the axons of the 12th cranial nerve. These nerves penetrate the mucosal lining and allow direct contact with the external environment without a peripheral sensory receptor relay. These nerves act as a chemical sensor, detect food scents, and play a role in social behaviors. Additionally, these nerves also offer a potential route for direct access of medication into the CNS [5,6,7]. Recent studies have shown the potential existence of a functional pathway (sometimes called “nose-to-brain” transport) for drugs to pass into the CNS from structures deep in the nose innervated by cranial nerves [5,7,8].

In this review, we focus on delivery of nanoformulations for the treatment of CNS disorders via the nasal route. This review includes a brief description of neurological diseases and currently developed nanoformulations suited for nose-to-brain delivery. We conclude with a brief discussion on the potential of nanomedicines and the future prospects of intranasal delivery to the CNS for successful clinical trials.

## 2. Blood Brain Barrier

The blood brain barrier (BBB) is made up of a triad of capillary endothelium, pericytes, and the astrocytic foot processes [9]. Specialized endothelial cells of the BBB lack fenestration, possess extensive tight junctions that severely restrict cell permeability, and have few pinocytic vesicles to minimize uptake of extracellular substances (Figure 1) [10]. Therefore, the transport of drugs is hindered and only few drugs can reach brain tissue. This is most likely why therapeutic agents that show in vitro efficacy fail to show in vivo activity. Drug delivery to the brain is always a great challenge unless the existing therapeutics have been customized [11]. It has been shown that small lipophilic molecules with molecular weights less than 400 Da can easily diffuse through the BBB, while large or hydrophilic molecules require special assistance like gated channels, proteins and/or the ligand-specific receptors, and ATP-mediated energy [12]. To maximize drug delivery into the brain, two basic approaches have been applied: a molecular approach and a polymeric carrier approach. In the molecular approach, drugs are delivered to the brain cells in the native form that are then activated by the target cell-specific enzymes. However, the limited availability of such drugs and their corresponding metabolic pathways has restricted the use of this approach. The polymeric carrier approach employs polymeric nanoparticles as the vehicles, which not only enhances the physiochemical stability of the therapeutic substances but also facilitates the administration through intravenous and intrathecal routes or as cerebral device implants [9].

## 3. List of CNS Diseases

### 3.1. Parkinson’s Disease (PD)

PD, occurring primarily in the substantia nigra, is the second-most common neurodegenerative disease, leading to the development of bradykinesia and tremors of cardinal motor functions (Figure 2) [3]. PD models specifically show a decrease in dopamine transporters, which are responsible for dopamine uptake by dopaminergic neurons and progression of neuronal communications. Reduced dopamine delivery during PD results in significant loss of neuronal functions [13]. Another hallmark of PD is the accumulation of α-synuclein in the Lewy bodies. However, the underlying mechanisms for PD-induced dementia are poorly understood [14]. The progression of PD can be delayed by levodopa—the precursor of dopamine—or a levodopa agonist [15,16]. However, it has been shown that untargeted delivery of levodopa can attack the peripheral nervous system leading to dyskinesia and adverse cardiovascular effects [17]. Therefore, it is advisable to carefully deliver neurotransmitters for PD treatment across the BBB by using a suitable delivery system, which does not allow penetration into other peripheral vessels.

### 3.2. Alzheimer’s Disease (AD)

AD is an irreversible and progressive neurodegenerative brain disorder that affects patients’ memory, cognition, language skills, and behavior [18]. At present, there is no known cure for AD given the lack of understanding of its molecular and intercellular mechanisms. However, treatments are available to improve the symptoms and on-going research is aimed at finding a cure. Accumulation of amyloid-beta (Aβ) peptides, neurofibrillary tangle formation of phosphoric tau proteins, and detrimental neuroinflammation are the signature features of AD progression [19]. The major biochemical markers for diagnostic purposes include Aβ plaques around the affected brains and presence of soluble Aβ and tau proteins in the cerebrospinal fluids, which are targeted. Inhibition of Aβ plaque/tau tangle formation and neutralization of their aggregations around neurons is the main focus of advanced therapeutic strategies [20,21,22]. The current clinically approved drugs can only relieve symptoms and delay AD progression by promoting interactions between neurons in AD brains through neurotransmitters [23]. Therefore, the discovery of novel AD markers and development of advanced and potent nanomedicines to target these AD markers is crucial for effective treatment and cure.

### 3.3. Glioblastoma (GBM)

GBM is the most common high-grade primary brain tumor in adults. It is also known to occur in children, albeit rarely [24]. GBM is the deadliest form of primary brain tumor, and originates from glial cells. It has been shown that even after aggressive multimodal therapy including chemotherapy, radiation, surgery, and their combined treatments, the median survival rate is 14 months, which emphasizes the urgent need for developing an effective strategy to eradicate brain tumors [1,25]. The discovery of highly expressed novel markers of brain tumors and the application of these markers in several nanoformulations functionalized with targeting ligands has notably improved the treatment and elimination of brain tumors [8,26,27,28,29,30,31]. Another recent and emerging anticancer approach that led to an explosion of clinical trials is immunotherapy, defined by targeting immune checkpoint receptors expressed on adaptive immune cells to improve immune surveillance [32]. However, with immunotherapy, the targeting efficiency is reduced resulting in incomplete elimination of tumors, and recurrence occurs because of the intratumor heterogeneity among individual patients. Therefore, the development of advanced and potent nanomedicines equipped with multimodal treatment systems is urgently needed for total removal of cancers.

### 3.4. Epilepsy

Epilepsy is a disorder of the CNS characterized by periodic loss of consciousness with or without epileptic seizures associated with abnormal electrical activity in the brain [33,34]. Epileptic seizures can result from almost any insult that perturbs brain function: for example, traumatic brain injury or stroke, infectious diseases such as neurocysticercosis, autoimmune diseases, and genetic mutations [35]. Currently, more than 500 genes associated with epilepsy have been identified. However, in most cases, epilepsy is idiopathic [33]. Epilepsy is related to physical risks and psychological and socioeconomic consequences which impair patients’ quality of life. An epileptic seizure is caused by abnormal excessive or synchronous neuronal activity in the brain. The epileptic seizure is a transient behavioral change, associated with symptoms like stiffening, jerking, loss of awareness, a smell of burnt rubber, or déjà vu, and a sensation that rises from the abdomen to the chest. Epileptic-seizure onset can be generalized (neuronal abnormal activity in a widespread distribution over both hemispheres), focal (neuronal abnormal activity in one or more localized brain regions or hemisphere), or of unknown origin (when it is not known whether the onset is focal or generalized) [36]. Onset of epilepsy is determined when there is >80% confidence based on the electroencephalography, clinical features, and neuroimaging findings [37].

### 3.5. Multiple Sclerosis

Multiple sclerosis (MS) is the well-known chronic demyelinating disorder of the CNS in young adults. MS is a heterogeneous, multifactorial, immune-mediated disorder, and is influenced by both genetic and environmental factors [38,39,40]. The initial stages of MS are characterized by reversible episodes of neurological dysfunction lasting several days or weeks. Irreversible clinical and cognitive deficits develop over time [39]. Some patients have a progressive disease course from the onset. Formation of demyelinating lesions in the brain and spinal cord is the pathological hallmark of MS, which can be associated with neuro-axonal damage [41]. During MS, infiltration of immune cells such as T cells, B cells, and myeloid cells into the CNS parenchyma causes focal lesions with associated injury [42]. MS creates a substantial burden on society with respect to high cost of available treatments, and poorer employment prospects.

### 3.6. Cerebral Palsy (CP)

CP is a common pediatric disorder occurring in about 2–2.5 per 1000 live births [43]. In the traditional sense, CP is not a disease entity, rather a clinical manifestation seen in children who share antenatal, perinatal, or early postnatal period-acquired features of a non-progressive brain injury or lesion. It has been noted that the clinical manifestations of CP vary greatly in the type of movement disorder, degree of functional ability and limitation, and the affected areas of the body [44]. Currently, there is no cure for CP, but there are advancements in areas of both prevention and treatment of brain injury. For instance, magnesium sulfate administration during premature labor and cooling of high-risk infants has been shown to reduce the rate and severity of CP [45]. Currently, most CP research and management strategies are focused on the needs of children, although the disorder affects individuals throughout their lifetime. Clinical researchers concerned with the management of children with CP are struggling to maximize function and participation in activities and minimize the factors that worsen the condition, such as feeding challenges, epilepsy, scoliosis, and hip dislocation [44]. Noteworthy management strategies comprise improving neurological function during early development, enhancing motor function through rehabilitation technologies, overcoming weakness and hypertonia, and preventing secondary musculoskeletal problems [44]. However, it is particularly challenging to meet the needs of people with CP in resource-poor settings.

## 4. Nanoformulations for Brain Disorders

### 4.1. Polymeric Nanosuspensions

Polymeric nanosuspensions are typical drug-loaded nanoformulations stabilized by using either lipid mixtures or non-ionic surfactants. Polymeric nanosuspensions have numerous advantages including enhanced drug loading, ease of fabrication, improved pharmacokinetics, and the possibility of surface modifications (Figure 3) [46]. However, the preparation of polymeric nanosuspensions takes very long and are not considered the formulations of choice for chronic disease therapy owing to their unstable shelf life [47].

### 4.2. Polymeric Nanogels

Polymeric nanogels are the cross-linked hydrophilic or amphiphilic polymers that are fabricated by emulsification followed by solvent evaporation [48,49]. The nanogel formulation is based on the principle of ionic and non-ionic polymers coalescing and forming cross-linked networks [50]. Polymeric nanogels are considered to provide more protection to the entrapped drugs during the transport process than other nanoformulations [49]. Polymeric nanogels have been mainly used to deliver DNA, siRNA, and oligonucleotides with an encapsulation efficiency of 40%–60% [51]. Nanogels have also been shown to deliver oligonucleotides specifically to the brain with more efficiency than to the spleen and liver [52].

### 4.3. Polymeric Nanoliposomes

Polymeric nanoliposomes represent a vesicular structure, containing an internal aqueous compartment and an outside covering of single or multi-lamellar lipid layers. This structural design of nanoliposomes facilitates enhanced stability and drug encapsulation along with evasion from the reticuloendothelial system [53]. Although the stability of nanoliposomes for brain disorders is still debatable, a report has shown that curcumin nanoliposomes were specifically active against amyloid aggregates [54].

### 4.4. Niosomes

Niosomes are nanoscale vesicles, having a stable bilayer structure, and are mainly composed of non-ionic surfactants and cholesterol. Niosomes are highly biocompatible and biodegradable [55]. They exhibit high chemical stability, long shelf life, low toxicity, and inexpensive manufacturing cost. Niosomes have the ability to entrap lipophilic or hydrophilic drugs and are able to deliver the drug molecules at target site in a sustained and/or controlled manner [56,57]. Niosomes have been reported to modify drugs organ distribution and metabolic stability [58]. It has been shown that surface modification of the niosomes promotes the target specificity for the cancer drug delivery system. For instance, modification of temozolomide-loaded niosomes with chlorotoxin, a target-specific peptide, significantly enhanced the gliomas targeting efficiency of the temozolomide [59]. A study reported that surface modified niosomes containing olanzapine (an atypical antipsychotic drug) showed a 3-fold increase in olanzapine concentration in the brain compared to the intranasal solution of the drug [60]. In an attempt to provide a novel pharmacological approach to ameliorate PD induced by subchronic MPTP administration in C57BL-6J mice, a group of researchers developed a non-invasive intranasal delivery system, composed of chitosan coated niosomes with entrapped pentamidine (inPentasomes). The study demonstrated that inPentasomes, because of their capability to inhibit glial-derived S100B activity, rescued the dopaminergic neuronal loss and reduced the severity of neuroinflammation occurred in the nigrostriatal pathway, which subsequently led to a significant improvement in parkinsonian motor dysfunctions [61]. Another similar investigation reported the preparation of drug free and pentamidine loaded chitosan glutamate coated niosomes for intranasal drug delivery using thin film hydration method. In this study, particular attention was given to observe the interactions of both drug free and pentamidine loaded niosomes with the mucin. It was demonstrated that niosomal formulation effectively delivered pentamidine or other possible drugs to the brain via nasal administration [62]. A study reported the formulation of buspirone hydrochloride (an anxiolytic agent and serotonin receptor agonist) niosomal in situ nasal gel in order to overcome the problems of short half-life (2–3 h) and low oral bioavailability (4%) of buspirone hydrochloride. It was shown that the application of niosomes proved the potential for intranasal delivery of the buspirone hydrochloride over the conventional gel formulations [63].

### 4.5. Nanospheres and Nanocapsules

Nanospheres are the solid core polymeric matrices fabricated by the micro-emulsion polymerization technique, whereas nanocapsules exhibit vesicular systems where a thin nontoxic polymer encapsulates the oil-filled drug compartment [64,65]. Both nanospheres and nanocapsules offer the advantages of improved drug stability, easy surface modification, and evasion of systemic degradation. However, they also have certain limitations such as complicated purification and storage as well as improper drug-release patterns [66,67]. Indomethacin-loaded nanocapsules have been shown to protect hippocampal cultures against in vitro inflammation [68].

### 4.6. Polymeric Nanomicelles

Polymeric nanomicelles contain a hydrophobic core surrounded by the shell constituting hydrophilic polymer blocks [69]. The shell stabilizes and disguises polymeric nanomicelles from cellular interactions, while the core can encapsulate about 30% of hydrophobic drugs [70,71]. It is estimated that polymeric nanomicelles could be effective for both in vitro and in vivo delivery of DNA molecules, although no reports exist yet on the nanomicelle-mediated CNS drug delivery. It has been shown in vitro that PEGylated phospholipid nanomicelles revoked amyloid-induced toxicity [72]. However, polymeric nanomicelles are not feasible for encapsulating hydrophilic drugs. Additionally, they have a shorter shelf life [73].

### 4.7. Metal Nanoparticles

Metal nanoparticles have been the focus of recent research given their potential applications in the fields of biomedical engineering and sciences [74]. Metal nanoparticles can be synthesized with the inclusion of several structural and surface modifications, which opens new horizons for their application in the fields of magnetic separation, targeted gene and drug delivery, and particularly, in diagnostic imaging [75,76,77]. Multiple modern and advanced imaging techniques like SERS, CT, MRI, PET, and ultrasound require a contrast agent for effective functioning. This requirement of a contrast agent provided the basis for the formulation of nano-sized gold, silver, and magnetic iron oxide (Fe_3_O_4_) nanoparticles [78,79,80].

### 4.8. Gold Nanoparticles

Gold nanoparticles (AuNPs) are widely used as nanomaterials for drug delivery and imaging [81]. Studies have shown that AuNPs exhibit low specificity because of absence of a selective moiety that can discriminate between targeted and non-targeted cells [82]. For delivering the therapeutic substances to targeted cells or tissues, researchers have been combined the AuNPs with cell-targeting ligands. The surface area of AuNPs provides a platform for conjugating multiple proteins, peptides, aptamers, and antibodies [83]. However, these conjugating methodologies are very complex as well as system-specific, which limits cross-system application. Additionally, several substances are not appropriate for clinical application because of potential toxicity resulting from the use of surfactants such as cetyl trimethylammonium bromide [84]. AuNPs for neuronal uptake can be utilized through two main routes: crossing the BBB and through the olfactory nerves. During an investigation, researchers successfully employed nose-to-brain direct transport pathway to deliver theranostic polyfunctional gold-iron oxide nanoparticles, surface loaded with miR-100 and antimiR-21, to GBMs in mice [85]. A study reported the formulation of resveratrol-loaded transferosomes and nanoemulsions labelled with gold nanoparticles for targeting the brain through intranasal route. The effectiveness of brain targeting of these two nanoformulations were achieved via testing the memory recovery of Wistar albino rats through a water maze test and bioaccumulation investigations using computed tomography and histopathological examination. It was observed that transferosomes significantly promoted behavioral acquisition and spatial memory function in the amnesic rats compared with both the nanoemulsion formulation and the pure drug [86]. It has been reported that AuNPs may lead to astrogliosis, increased seizure activity, and judgement impairments after crossing the BBB [87].

### 4.9. Silver Nanoparticles

Silver nanoparticles (AgNPs) have been shown to induce cytotoxicity in human skin, lungs, and fibroblast cells [88,89]. In relation to the CNS, it has been shown that AgNPs, following inhalation and ingestions, cross the BBB and accumulate in the brain [90,91,92]. Patchin et al. found rapid translocation of 20 nm AgNPs into the olfactory bulb, with slower and less effective transport of 110 nm silver particles after a 6-h exposure [93]. A study reported very little AgNP absorption (measured as total silver) into the blood after intranasal administration and significantly higher blood concentrations after AgNO3 delivery, and demonstrated that silver found in the blood was due to silver ion release from AgNPs [94]. AgNPs have also been shown to induce cytotoxicity in neurons in vitro [95]. However, the exact mechanisms of AgNPs causing neurodegeneration are poorly understood, and this topic needs to be more thoroughly investigated. In contrast to the above facts, a study reported that AgNPs showed remarkable anti-inflammatory effects, reduced LPS-induced ROS, nitric oxide and TNFα production, which resulted into decreased microglial toxicity towards dopaminergic neurons [96]. Therefore, further investigations are required to take decisions about how to design future classes of safe AgNPs.

### 4.10. Magnetic Nanoparticles

Magnetic nanoparticles (MNPs) are actually the nanoparticles which exhibit magnetic properties. These are capable of producing temporary pores in the cell membranes, as is the case in the BBB endothelium, which enhance drugs targeting and delivery; the phenomenon is termed as magnetoporation [97]. MNPs have been utilized in multiple biomedical applications including magnetic hyperthermia and heating, magnetic vectors and magnetic contrast agents [98,99,100]. In an attempt to establish a promising treatment for PD, researchers developed a nanocarrier composed of Fe3O4 nanoparticles coated with oleic acid molecules and absorbed short hairpin RNA. It was shown that these superparamagnetic nanoparticles reduced the expression of α-synuclein, suppressed its toxic effects on the cells, and blocked α-synuclein-induced cell death [101]. Another study reported the successful delivery of mesenchymal stem cells (MSCs) and improved neurobehavioral assessment when MSCs were incubated with micrometer-sized iron particles and finally administrated them in a PD mouse model by the way of the intranasal route [102]. It has been recently demonstrated that dextran-coated iron oxide nanoparticles enhanced the therapeutic efficacy of human MSCs in a mouse model of PD by decreasing the loss of dopaminergic neurons and increasing the differentiation of human MSCs to dopaminergic neurons [103].

### 4.11. Dendrimers

Dendrimers have a characteristic architecture consisting of molecular hooks and are a novel class of highly branched nanoparticles that can target specific cells [104]. Two basic structures for dendrimers have been demonstrated of which one represents a central core with radiating polymer branches, and the other type only shows multiple branches without the core [105]. Because of the unique branching structure of dendrimers, surface modifications through either adsorption or covalent conjugation become very easy, which also enhances the potential of dendrimers to carry various drugs [106,107]. Polyamidoamine dendrimers have been shown to be used to fabricate tunable drug delivery systems with the potential to target intracellular components both in vitro and in vivo [108]. Additionally, dendrimers can also be used as scaffolds for delivering therapeutic and diagnostic entities in vivo.

## 5. Limitations of the Existing Routes of Administration

Although nanotherapeutics has shown tremendous application potential, it still has some limitations [109,110,111,112,113,114,115,116,117,118,119,120,121,122,123,124,125,126,127,128,129,130,131,132,133,134,135,136,137,138,139,140,141,142,143] (Table 1). The brain is the most sensitive and complex organ and any non-specific distribution of drugs may result in complicated irreversible damage to the CNS. Nanoparticles are very small in dimension and could likely deliver drugs way over the tolerated levels by the brain, thereby delaying the clearance and resulting in severe toxic effects [144]. Therefore, intensive care is needed when considering the delivery of therapeutics using nanoparticles. The toxicity of nanoparticles relates to their shape, size, surface area, solubility, and dose, and oxidative stress generation is the most commonly reported toxicity [145]. It has been shown that iron oxide nanoparticles reduced the viability of the PC12 cell line, whereas they caused neuronal degeneration in vivo [146,147]. A study reported the non-specific distribution of silver nanoparticles in the brain, liver, and kidney upon long-term exposure [148]. To overcome these limitations, biodegradable polymeric nanoparticles are preferred these days which can offer easy drug delivery and surface modifications. It has been shown that the biodegradable polymeric nanoparticles are metabolically converted to biocompatible lactic acid, butanol, and 6-hydroxycaproic acid, all of which are considered safe by the US-FDA [149].

## 6. Nose-to-Brain as an Alternate Therapeutic Route

Although researchers have successfully explained active transport mechanisms from the blood into the CNS and enabled the BBB penetration of some drugs, additional challenges still exist. Several drugs, particularly macromolecules, are degraded in the gastrointestinal tract and/or undergo hepatic metabolism, which severely limits the bioactive drug reaching the blood stream [150,151]. It is not appropriate to increase the oral dose for compensation, because it leads to unacceptable gastrointestinal tract or systemic adverse effects. Generally, 100% bioactive drugs in the blood stream can be obtained via injections, but they are not the route of choice in many cases especially for treatment requiring frequent dosing or home administration [152]. This limitation is even more conspicuous for intrathecal drug administration. Hence, researchers have focused on nasal administration that can offer an alternate route into the systemic circulation for multiple drugs having low oral bioavailability, slow absorption, and slow onset of action (Figure 4) [153]. In this scenario, the nasal passage can be used as an attractive delivery route for CNS drugs that do pass the BBB. Moreover, it can also be used for drugs formulated to exploit active transport mechanisms to cross from the blood into the brain [154].

## 7. Formulations for Nose-to-Brain Delivery

Regarding the nose-to-brain delivery, various clinical and preclinical studies have been conducted to involve the use of drugs in solution and particulate dispersions [155,156,157]. Clinical studies have usually involved the use of a nasal drug delivery device, while most animal studies have been conducted in rodents.

### 7.1. Solution Dosage Forms

Studies have reported the use of drug molecules via the nose-to-brain route by simply dissolving it in an aqueous phase which produced significant pharmacological effects [155,157,158,159]. Intranasal delivery of insulin to the brain in an insulin solution was among the first trials of delivering peptides to the CNS [160]. Although clinical studies have shown a pharmacological response of drugs administered via the nose-to-brain route, preclinical studies reveal that only a small fraction of the administered dose is actually delivered to the brain. A study reported the delivery of radiolabeled interferon beta-1b in aqueous solution form to a monkey brain with a Cmax of 0.0064%, and concluded that the drug delivery would be enhanced by adding absorption enhancers to the formulation [157]. Brain weight was supposed to be 1% of the animal’s average body weight in all cases where the Cmax was shown as a percentage of the total dose [161]. A midpoint is considered as the representative body weight when a range of body weights are given. Another study reported the delivery of oxytocin solution to the brain via the nasal route with a Cmax of 0.003% of a 10 μg dose being found in the brain [162]. Wang et al. delivered the DB213 (HIV replication inhibitor) solution to the rat brain with a Cmax that was estimated to be no more than 0.007% of the administered dose [163]. In comparison with similar computations following oral dosing where the Cmax is 0.24%–4.3% of the administered dose, these Cmax values are extremely low [164]. However, the addition of specialized excipients to these solutions boosts brain delivery via the nasal route. For example, when tetradecyl-b-d-maltoside (a penetration enhancer) was added to the protein solutions of serpin B2 and activin administered via the nose-to-brain route, neuroprotective activity was observed in a mouse model of brain injury [165]. A study showed that addition of a cell-penetrating peptide (CPP), l-penetratin, to the exendin-4 (glucagon-1 receptor agonist) solution resulted in delivery of exendin-4 to the hypothalamus and hippocampus on nasal delivery to normal mice and the activation of insulin signaling, whereas plain exendin-4 solution and exendin-4 plus the inactive d-penetratin did not show brain delivery [166]. Moreover, intranasal exendin-4/CPP solutions plus supplemental insulin resulted in a therapeutic response against severe cognitive dysfunction in an SAMP8 mouse model of accelerated senescence [166].

Researchers have recommend the use of viscosity-building agents such as carboxymethylcellulose to enhance the nasal residence time of nasal solutions, and thus increase drug transport through the olfactory neurons [167]. It has been shown that when methotrexate solution containing carboxymethylcellulose was administered through the nasal route in combination with oral acetazolamide in a rat 9 L glioma model, significant tumor repression was observed when compared to drug delivery via an intraperitoneal route [167].

### 7.2. Nanoparticles for Nose-to-Brain Delivery

With conventional nasal solutions, very low drug transfer levels have been observed. Therefore, to address the low drug delivery problem, scientists are conducting experiments with nanoparticulate formulations like nanoemulsions, lipids, or polymer particles, which offer enhanced penetration and a longer residence time within the nasal cavity [156,168,169,170,171,172,173,174,175,176] (Table 2). It has been found that 100 nm nanoemulsion particles penetrated the olfactory bulb and reached the brain to a small extent, whereas 900 nm particles could not penetrate the brain, which indicated that a particle size cut-off may be operational for the delivery of nanoformulations beyond the olfactory bulb [168]. A transformational effect is known to occur on the level of drugs detected in the brain following intranasal delivery when it is converted from solution to particulate formulation [177]. The solution of a delta selective opioid agonist, leucine-5-enkephalin (LENK), initially failed to reach the rat brains in a considerable amount via the nasal route. However, the delivery was increased when LENK was formulated as an absorption-enhancing chitosan-based nanoparticle [156]. Compared to an intranasal dose of rivastigmine (a cholinesterase inhibitor used for the treatment of dementia) in solution, the emulsion form when administered intranasally led to a 5-fold increase in brain exposure [178]. The intranasal delivery of an anti-psychotic drug, quetiapine, resulted in a 2.57-fold increased Cmax when administered as chitosan-tripolyphosphate nanoparticles instead of the conventional solution [179]. From the commercial point of view, solution-based formulations exhibit a short half-life and are more prone to microbial contamination. Nanoformulations for nasal drug delivery can be further divided into solid lipid nanoparticles and nanoparticles prepared from chitosan derivatives or poly(l-lactide-co-glycolide), as described below [156,180,181,182].

### 7.3. Lipid Nanoparticles

Lipid nanoparticles comprise a lipid core stabilized by a surfactant. Lipid nanoparticles differ from oil-in-water emulsions in that they are solids at room temperature. They are prepared by melting the lipid, followed by size reduction and surfactant stabilization of the resulting particles in an aqueous disperse phase [192]. Lipid nanoparticles can be loaded with hydrophobic drugs and may be administered via the nasal route to deliver drugs to the brain. Valproic acid lipid nanoparticles, compared to the drug in solution form, delivered high drug dose to the brain, and prevented tonic-clonic partial seizures in a maximal electric shock seizure model; this effect was similar to intraperitoneal phenytoin [180]. The lipid nanoparticles are supposed to protect the drug from biological and/or chemical degradation, and from extracellular transport by P-gp efflux proteins, and may indeed promote drug transport by unknown mechanisms [193].

### 7.4. Microemulsions and Nanoemulsions

Microemulsions (MEs) are pseudo ternary systems which consist of oil, water and surfactant. They are frequently used in combination with co-surfactants. MEs are stable, single-phase swollen micellar solutions which form spontaneously, and can be utilized to incorporate a larger quantity of hydrophilic and/or lipophilic drug molecules [183]. Nanoemulsions (NEs) comprise of mixtures of oil, water and surfactant. These are kinetically stable, non-equilibrium systems, which do not essentially require the co-surfactants. NEs synthesis is not spontaneous but requires high energy input. NEs are also called mini-emulsions, submicron emulsions, or ultrafine emulsions [184]. MEs and NEs share similarity in size, however, they are different from each other regarding the method of preparation, composition, and concentrations of the components. It has been suggested that MEs and NEs enhance nose-to-brain delivery of drugs through the olfactory region [185,186]. Additionally, they offer dose uniformity and better sprayability advantages over other nano vectors for intranasal administration. It has been reported that compared to intravenous administration, ME of nimodipine (a calcium channel blocker) showed higher nimodipine concentration in the brain following intranasal administration [187]. A study has shown that olanzapine mucoadhesive NE resulted in high brain to plasma ratio than olanzapine nanoemulsion and olanzapine following intranasal administration [186]. Additionally, Vyas TK and co-workers reported similar findings with sumatriptan, clonazepam, resperidine and zolmitriptan MEs [188,189,190,191]. These studies suggest that direct nose-to-brain delivery of MEs and NEs may provide significant merits in many therapeutic situations where rapid and/or specific targeting of drugs to the CNS is required.

### 7.5. Nanoparticles Composed of Chitosan and Chitosan Derivatives

Chitosan has been shown to act as a penetration enhancer by temporarily opening intercellular tight junctions, and has been utilized in a number of nose-to-brain nanoformulations [194]. Although various reports have shown the enhanced delivery of drugs using chitosan nanoparticles, the exact mechanism of this enhanced drug delivery remains poorly understood [179]. Compared to an intranasal solution, quetiapine chitosan nanoparticles enhanced the drug delivery to the brain by 34% [179]. It has been reported that intranasal chitosan nanoparticles of pramipexole better controlled motor deficits in a rotenone model of PD than the oral or solution dosage form of the drug [195]. Additionally, chitosan nanoparticles have also been shown to be used in gene therapies via nasal administration. For instance, chitosan-tripolyphosphate siRNA nanoparticles silenced galectin-1, a gene that drives chemoresistance and immune-therapy resistance on intranasal administration, resulting in increased survival in a mouse tumor model [181]. Chitosan nanoparticles prepared with a chitosan-mangafodipir (mangafodipir is a manganese dipyridoxyl diphosphate chelator) electrostatic complex showed effective delivery of siRNA to the olfactory bulb for gene silencing [196,197].

The delivery of nucleic acids to the brain via the nasal route is a breakthrough achievement. A chitosan amphiphile has been shown to deliver a labile peptide to the brain. When *N*-palmitoyl-*N*-monomethyl-*N*,*N*-dimethyl-*N*,*N*,*N*-trimethyl-6-*O*-glycolchitosan (Nanomerics’ Molecular Envelope Technology-MET) nanoparticles encapsulating LENK were administered intranasally, it produced analgesia in all tested animals with exclusive central activity and no peptide detected in the periphery [156]. A study has reported that intranasal administration of MET-propofol formulations produced sedation in a healthy rat model [198]. Furthermore, MET is mucoadhesive, which prolongs the residence time of the formulation within the nares, leading to extended duration of drug action; however, it does not open tight junctions [164]. MET is also a penetration enhancer, and has shown enhanced penetration in the gut epithelium via particle uptake mechanisms [199,200]. Chitosan-based emulsions have been shown to improve drug deposition in the brain following intranasal delivery. For instance, 0.3% (*w*/*v*) chitosan significantly increased the brain deposition of zolmitriptan when administered in an oil-in-water emulsion [201]. It has been reported that when resveratrol lipid microparticles were coated with chitosan and administered intranasally as lipid particles at the dose of 60 μm to a rat model, a dramatic and specific 6-fold increase, without any detectable systemic exposure, was observed in its distribution to the cerebrospinal fluid [202].

### 7.6. Poly l-Lactide-co-Glycolide (PLGA) Nanoparticles

PLGA is a globally approved polymer for use in the human drug delivery system because of its non-toxic and biodegradable nature [203]. PLGA may be used to prevent degradation of drugs in the nasal cavity and may be loaded with hydrophobic drugs; these properties have been exploited in nose-to-brain delivery [203]. It has been shown that olanzapine PLGA nanoparticles resulted in 10-times more Cmax and drug delivery to the brain than olanzapine solution [182]. A study reported that oxcarbazepine PLGA nanoparticles showed better pharmacokinetic behavior and superiorly reduced intraperitoneal pentylene tetrazole-induced seizures in a rat model than the drug in solution form [204]. It was shown that upon intranasal administration, a PLGA-poly(ethylene glycol) (PLGA-PEG) copolymer nanoparticle, conjugated with *Solanum tuberosum* lectin and loaded with basic fibroblast growth factor, improved cognition in a mouse AD model [205]. Interestingly, although PLGA nanoparticles have not been reported to be mucoadhesive or penetration enhancers, drug delivery to the brain is still enhanced through the nasal route. In a trial to coat the PLGA nanoparticles with chitosan, it was observed that their brain transport was altered and positively charged chitosan-coated PLGA nanoparticles appeared to move slower than plain negatively charged PLGA nanoparticles from the caudal to the rostral regions of the brain [206]. Another study developed huperzine A-loaded, mucoadhesive, and targeted PLGA nanoparticles with surface modification by lactoferrin-conjugated *N*-trimethylated chitosan for efficient intranasal delivery of huperzine A to the brain for AD treatment. These nanoparticles showed good sustained-release effect, adhesion, targeting ability, and a broad application prospect as a nasal drug delivery carrier [207]. These studies suggest that particle transport via the nose-to-brain route is highly affected by the particle surface chemistry. Furthermore, clarification of the different biological mechanisms of nose-to-brain delivery will assist in the design and development of various useful dosage forms.

## 8. Limitation and Safety Consideration for the Nasal Formulations

The intranasal-route of drug delivery is an attractive route which quickly and accurately accesses the brain. The intranasal route of drug delivery has multiple advantages such as BBB evasion; being non-invasive, convenient, and a patient-friendly route of drug administration; having faster onset of action, more precise drug targeting, more significant area of drug absorption; circumventing the hepatic first-pass metabolism of drug; and showing less systemic side effects [208,209,210]. However, the clinical application of intranasal formulations for brain drug delivery still has a long way to go. Poor drug permeability from the nasal mucosa, enzymatic degradation of the drug, mucociliary clearance, low drug retention time, and nasomucosal toxicity are some of the common limitations of the intranasal drug delivery [211,212]. Various controlled delivery systems, colloidal drug carriers, permeation enhancers, and other novel approaches have been employed to improve the drug permeability and absorption [213,214]. It has been shown that the use of a suitable mucoadhesive system like mucoadhesive polymers, viscous formulation, in situ gelatins, and hydrogel enhances the retention time and reduces mucociliary clearance [215]. Additionally, some protective measures are needed, like encapsulation in a nanocarrier system, which prevent enzymatic degradation of the drug. All these formulation strategies facilitate intranasal drug delivery; however, the clinical success of intranasal therapy remains limited because of the high and frequent dose of the formulation, hence irritating the nasal mucosa. Additionally, the protective barriers of the nasal mucosa limit the efficiency of intranasal therapy, and only 1% or <1% of the drug reaches the brain after intranasal administration. Thus, it is essential to focus on the development of a suitable formulation to overcome these barriers [216].

Furthermore, the nature and efficacy of the drug and excipients should also be considered. Compared to the other routes, the nasal cavity allows only a small amount of formulation (100–200 μL) at a time given its relatively low volume (25 cm^3^). Hence, a potent drug is required for intranasal drug delivery to the brain. Moreover, it is very important for excipients in a formulation to be biocompatible and not produce any aggressive odor [217]. Additionally, the tonicity, viscosity, and pH (5.0–6.5) of the formulation also play key roles in drug development [216,218]. Another key factor to be considered is the technique of administration that influences drug absorption by the brain. The formulation is prone to mucociliary clearance if it is deposited on the floor of the nasal cavity. It has been shown that the posterior and upper regions of the nasal cavity are responsible for drug absorption to the olfactory region or the brain, whereas the anterior region of the nasal cavity tends to displace the drug towards systemic circulation. Hence, a suitable delivery device such as a nasal dropper, needleless syringe, or spray is required to deliver the formulation in the appropriate region of the nasal cavity [219]. The currently available devices to target drugs to the brain are OptiMist™ (a breath actuator) and ViaNasa™ (electronic atomizer) [216,220,221].

Many researchers have claimed successful, direct, and effective drug transfer from the nasal cavity to the brain although some reports have contradicted the hypothesis of direct drug delivery to the brain. Scientists from Leiden University did not find any evidence of direct nose-to-brain delivery of melatonin, estradiol, and vitamin B12, whereas another group of researchers reported significant amounts of these drugs in the brain after intranasal administration [222,223]. This discrepancy in results is likely because of the experimental conditions and formulation factors, as well as variable methodology used. Hence, it is essential to thoroughly understand all the formulation aspects for successful clinical application of intranasal drug delivery to the brain [222]. Despite significant successful research work, intranasal drug delivery requires more efforts for commercial availability of these drugs. Recent and on-going research has only focused on few issues, whereas the development of successful formulations requires an all-round and deep consideration and understanding.

## 9. Future Prospective of Nose-to-Brain Delivery

Treatment of neurological diseases remains one of the most significant challenges, and advances in nanotechnology have provided promising solutions to this challenge [224]. Based on the past few years’ research, we can conclude that nanotechnology has gained considerable focus. Multiple nanocarriers such as solid lipid nanoparticles, liposomes, polymeric nanoparticles, dendrimers, nanogels, micelles, nanoemulsions, and nanosuspensions have been studied for the delivery of brain therapeutics [224]. It is expected that in the near future, more drugs in the form of nasal formulations intended for brain disorders will be commercially available [225]. However, this functional drug delivery mechanism to the brain is a potential area of research because there are still certain unresolved challenges during intranasal delivery. These include handling large molecular weight polar drugs such as peptides and proteins, low membrane permeability, mucocilliary clearance, and the possibility of an enzymatic degradation of the molecule in the lumen of the nasal cavity. These problems can be solved by focusing on bioadhesive excipients and absorption enhancers in the formulation. The current nanoparticle-based drug delivery technology should be improved further, so that it can be target oriented, safe, effective, and cost-effective. Additionally, development of CNS nanoformulations needs to focus on improving their BBB permeability, reducing neurotoxicity, and increasing their drug-trafficking performance and specificity for brain tissue using novel targeting moieties [224]. Furthermore, adequate clinical and preclinical trials to improve the intranasal delivery system are required. It is also not entirely clear how drugs are delivered directly to the brain; thus, further research is required to better understand the exact mechanism of drug passage through the intranasal route to specific brain areas. It is also important to pay attention to formulation strategies, drug delivery devices, new excipients development, and mucoadhesive characteristics of polymers, all of which could potentially improve bioavailability, prolong retention, and maximize the effects of the drugs. Additionally, toxicodynamic studies of drug and excipients and nanotoxicity of nanocarriers should also be extensively investigated [225].

## 10. Conclusions

Drug carriage and accessing the brain has always remained a significant challenge in the treatment of CNS disorders. The efficiency of CNS drugs becomes limited due to various physiological factors such as first pass effect, enzymatic degradation, presence of the BBB, inadequate blood perfusion, systemic clearance, peripheral side effects, and reduced bioavailability. The intranasal route offers many advantages and can hence overcome some of the limitations; it is thus a preferred, alternative drug administration route over the parental and oral routes. Currently, scientists have utilized different novel strategies such as targeting ligands, nanoparticulate systems, and mucoadhesive formulations to develop a promising intranasal drug delivery device with minimal toxicity and side effects. Most of the investigations are currently in preclinical or early clinical stages, and the successful claims are limited to animal models only. Very few investigations have expanded to human clinical trials; however, it is estimated that the intranasal route could be a future method for drug delivery to the brain. Our review has discussed the various scientific attempts in the development of an effective intranasal drug delivery system for the treatment of brain disorders. A large number of drugs, proteins, peptides, biological agents, and cells are presently under investigation for intranasal delivery awaiting successful outcomes. If clinical studies support such preclinical data, intranasal drug delivery can be a new beacon of hope for treatment of brain disorders.

## Figures and Tables

**Figure 1 molecules-25-01929-f001:**
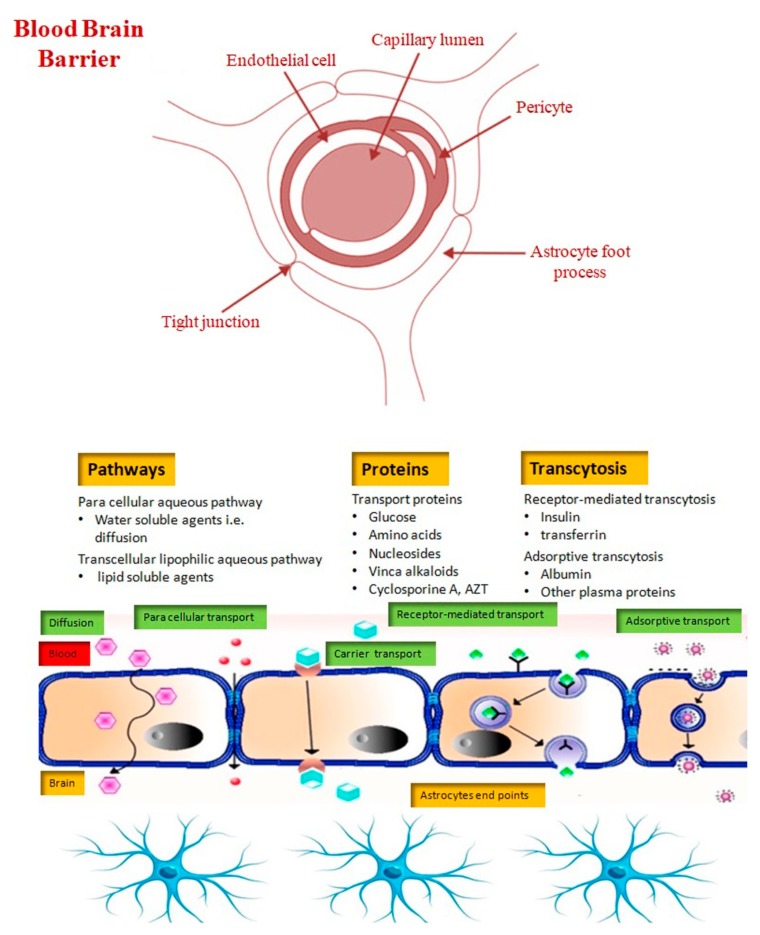
Schematic demonstrating various transport systems that shuttle molecules across the BBB. Very small amount of water-soluble compounds cross through the tight junctions (paracellular), whereas lipid-soluble agents traverse via the transcellular lipophilic pathway. Selective transport systems exist for glucose, amino acids, nucleosides, and other substances, in addition to specific receptor-mediated endocytosis for certain proteins such as insulin and transferrin. (AZT = azathioprine).

**Figure 2 molecules-25-01929-f002:**
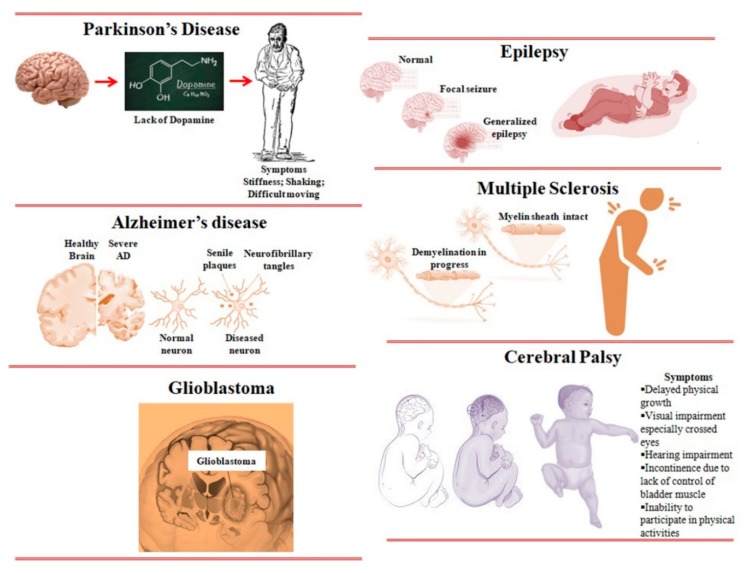
Important neurological disorders.

**Figure 3 molecules-25-01929-f003:**
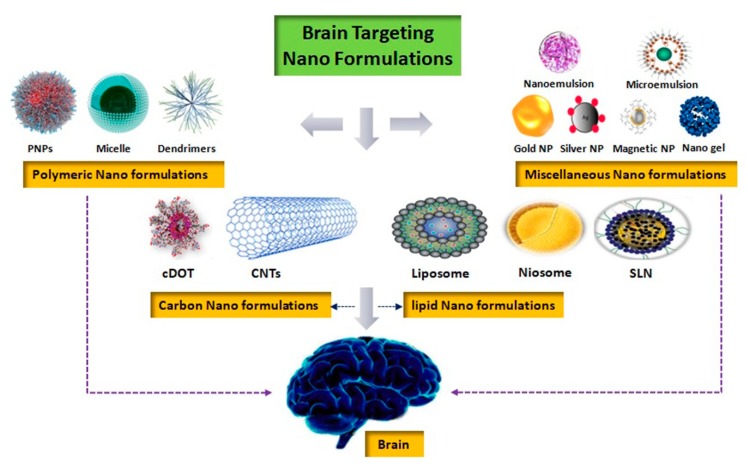
Various important brain targeting nanoformulations.

**Figure 4 molecules-25-01929-f004:**
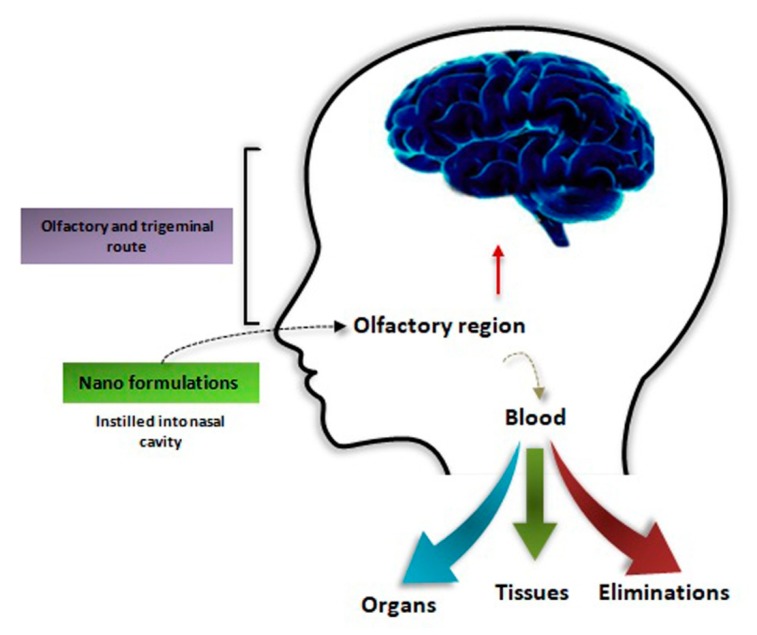
Brain targeting pathways in intranasal administration.

**Table 1 molecules-25-01929-t001:** Approaches for brain drug delivery.

S. No	Approaches	Benefits	Drawbacks	References *
1	Nanoparticles	Target the brain using specific physiological conditions; Actively targeted drug delivery	Cross the BBB	[114,115,116,117,118,119,120,121,122,123,124]
2	Gold nanoparticles	Drug delivery systems, x-ray imaging, photothermal and photodynamic therapies	Neurotoxic effects like astrogliosis, increased seizure activity, and judgement impairments	[81,82,83,84,85,86,87]
3	Silver nanoparticles	Drug delivery systems, anti-inflammatory	Neurotoxic	[88,89,90,91,92,93,94,95,96]
4	Magnetic nanoparticles	Targeted drug/gene delivery, contrast agents for MRI, biosensors for diagnostic purpose, hyperthermia as treatment modality in cancer	Insufficient size control distribution, uncontrolled shape, poor colloidal stability, nonbiodegradability, limited biocompatibility and cytotoxicity	[97,98,99,100,101,102,103]
5	Nanoparticles for brain diagnostics or imaging	Cross the BBB through increasing the permeability under diseased states; Enhanced imaging	Difficult understanding of dynamic changes in the BBB, Cross the BBB	[143]
6	Brain permeability enhancers	Open the BBB transiently	Mismatched results between humans and rodents	[128,129,130,131]
7	Enhanced brain drug uptake using non-invasive techniques	Ability to open the BBB and reduce efflux transporters	Higher toxicity	[136]
8	Viral vectors	High transfecting efficiency of genes	Safety issues; direct injection to brain; crossing the BBB; high dose by intravenous route	[109,110,111,112,113]
9	Exosomes	Delivering the genes to CNS; actively cross the BBB	Difficult loading procedure; require exosomes donor cells; in vitro toxicity, poor pharmacokinetics	[125]
10	Niosomes	Targeted drug delivery, reduced dose is required, subsequent decrease in side effects, improved bioavailability, osmotically active and stable	Requires specialized equipment, inefficient drug loading, time consuming	[55,56,57,58,59,60,61,62,63]
11	Delivery via active transporters in the BBB	Potently cross the BBB by intravenous injection	Used for small molecules only	[126,127]
12	Delivery under disease states through permeable BBB	Potentially cross the BBB	Dynamic changes in the BBB and their mechanisms are poorly understood	[133,134,135,137,138]
13	Using altered administration routes	Bypass the BBB through nasal administration	Suitable for low dose only	[142]

* The numbers refer to the numbered references in the text.

**Table 2 molecules-25-01929-t002:** List of nanoformulations for intranasal drug delivery, with their potential advantages and limitations.

S. No	Nanoformulation	Advantages	Limitations	References *
1	Polymeric nanoparticles	Higher loading efficiency	Biocompatibility issues	[170]
2	Solid lipid nanoparticles	better control upon drug release pattern; Improved bioavailability of incorporated drug molecules	Unpredictable gelatin tendency and particle growth	[173]
3	Microemulsions and nanoemulsions	Thermodynamically stable; increased rate of absorption; enhance bioavailability	Stabilization of nanoemulsions require large concentration of surfactants as well as high energy input	[169,183,184,185,186,187,188,189,190,191]
4	Nanostructured lipid carriers	Non-toxic; high loading capacity; controlled and targeted release	Issues with physical stability	[174]
5	Polymeric micelles	Low toxicity; High stability; High dose loading	Immature drug-entrapping technology; complicated polymer synthesis	[176]
6	Dendrimer-conjugate nanoparticles	Better biodistribution and pharmacokinetics; targeted, site specific and controlled drug release	Toxic	[174]
7	Polymer-lipid hybrid nanoparticles	Targeted delivery; minimum side effects; sustained release drug; low frequency of administration	Storage and stability issues	[175]
8	Chitosan nanoparticles	Non-toxic; stable; biodegradable; biocompatible; enhanced absorption	Time consuming protocols of synthesis; need organic solvents in preparation method	[172]
9	PLGA nanoparticles	Minimum toxicity; deeper penetration into the tissues; high loading capacity; extended drug release	Toxicity issues	[171]

* The numbers refer to the numbered references in the text.

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
