# Peer review of "Intranasal Delivery of Nanoformulations: A Potential Way of Treatment for Neurological Disorders"

_molecules, 2020, doi:10.3390/molecules25081929_

Round 1
Reviewer 1 Report
This paper deals with the positive and negative aspects of nose-to-brain delivery of nanoformulations for treatment of multiple brain disorders
Some major points have to be successfully addressed before publication.
The authors seem to list nanoformulation for N2B without weighing up the pros and cons, as reported in the title.
Moreover the review does not cover the literature very widely, especially the more recent one.
Few examples:
some nanoformulations, such as nanoemulsions, niosomes, micro/nanobubbles, are not cited;
references 169 and 204 are reviews on nanoparticulate systems for brain delivery published in 2001; 48 review articles have been published in 2019-2020 on the same topic (source Scopus);
ref 215 is a review article on nasal drug delivery published in 2009; 79 review articles have been published in 2019-2020 on the same topic (source Scopus).
Author Response
This paper deals with the positive and negative aspects of nose-to-brain delivery of nanoformulations for treatment of multiple brain disorders
Some major points have to be successfully addressed before publication.
>The authors seem to list nanoformulation for N2B without weighing up the pros and cons, as reported in the title.
Response: The title of the article has been modified as suggested by the reviewers. Additionally, the advantages and the limitations of all the nanoformulations have been summarized in the Table 1 and Table 2 along with the appropriate references.
>Moreover the review does not cover the literature very widely, especially the more recent one.
Few examples:
some nanoformulations, such as nanoemulsions, niosomes, micro/nanobubbles are not cited;
Response: Section 7.4. (Microemulsions and nanoemulsions) and section 4.4. (Niosomes) have been added, and also described in the Table 1 and 2. Micro/nanobubles have not been described in this article as they are related to “Ultrasound-responsive drug delivery systems (URDDS)” and do not come in the scope of our article.
>references 169 and 204 are reviews on nanoparticulate systems for brain delivery published in 2001; 48 review articles have been published in 2019-2020 on the same topic (source Scopus);
>ref 215 is a review article on nasal drug delivery published in 2009; 79 review articles have been published in 2019-2020 on the same topic (source Scopus).
Response: We do agree with editorial and reviewer comments but we wish to bring in your kind notice that our review summarizes the recent and relevant knowledge regarding the nanonformuatlions delivery through nose to brain for the treatment of various neurological disorders. Our study is based on the recent and the most relevant findings covering wide range of topics including treatment of neurological diseases with nanonformulations, last but not the least the limitations and future prospects of nose to brain drug delivery system as a future treatment remedy for various complicated brain disorders.
Reviewer 2 Report
The manuscript intituled ¨The positive and negative aspects of nose-to-brain delivery of nanoformulations for treatment of multiple brain disorders¨, is very well written and its approach is interesting. I would like to do a few recommendations before the final decision.
- The title should be more appropriate with the content of the manuscript.
- The manuscript should include gold, silver, and magnetic nanoparticles in Tables 1 and 2.
- It's important to add information about these particles (gold, silver, and magnetic nanoparticles) with respect to their toxicity by this route. Moreover, add these systems separately in each topic in the manuscript.
- The review is extensive. Despite this, microemulsions have not been particularly discussed. Only in the citation 177 therefore I think that they should be mentioned independently, since there is a bibliography that mentions their viability. See next article:
Development of a Nasal Donepezil-loaded Microemulsion for the Treatment of Alzheimer ’s disease: in vitro and ex vivo Characterization. Author(s): Lupe C. Espinoza, Marisol Vacacela, Beatriz Clares*, Maria Luisa Garcia, Maria-Jose Fabrega, Ana C. Calpena. Journal Name: CNS & Neurological Disorders - Drug Targets (Formerly Current Drug Targets - CNS & Neurological Disorders). Volume 17 , Issue 1 , 2018. DOI : 10.2174/1871527317666180104122347
Author Response
The manuscript intituled ¨The positive and negative aspects of nose-to-brain delivery of nanoformulations for treatment of multiple brain disorders¨, is very well written and its approach is interesting. I would like to do a few recommendations before the final decision.
>The title should be more appropriate with the content of the manuscript.
Response: Title has been revised.
>The manuscript should include gold, silver, and magnetic nanoparticles in Tables 1 and 2.
It's important to add information about these particles (gold, silver, and magnetic nanoparticles) with respect to their toxicity by this route. Moreover, add these systems separately in each topic in the manuscript.
Response: Sections 4.8 (Gold nanoparticles), 4.9 (Silver nanoparticles), and 4.10 (Magnetic nanoparticles) have been added. These nanoparicles have also been included in the Table 1.
>The review is extensive. Despite this, microemulsions have not been particularly discussed. Only in the citation 177 therefore I think that they should be mentioned independently, since there is a bibliography that mentions their viability. See next article:
Development of a Nasal Donepezil-loaded Microemulsion for the Treatment of Alzheimer ’s disease: in vitro and ex vivo Characterization. Author(s): Lupe C. Espinoza, Marisol Vacacela, Beatriz Clares*, Maria Luisa Garcia, Maria-Jose Fabrega, Ana C. Calpena. Journal Name: CNS & Neurological Disorders - Drug Targets (Formerly Current Drug Targets - CNS & Neurological Disorders). Volume 17 , Issue 1 , 2018. DOI : 10.2174/1871527317666180104122347
Response: Section “7.4. Microemulsions and nanoemulsions” has been added. Additionally, the suggested reference has been cited as Ref# 179.
Round 2
Reviewer 1 Report
The paper seems to be successfully improved.
Some minor points have to be still addressed before publication.
Authors added a section on niosomes but none of the cited references (55-59) are related to niosomes and nose-to-brain
Few examples of recent studies:
inPentasomes: An innovative nose-to-brain pentamidine delivery blunts MPTP parkinsonism in mice.
Rinaldi F et al J Control Release. 2019 Jan 28;294:17-26. doi: 10.1016/j.jconrel.2018.
Formulation and Evaluation of Niosomal in situ Nasal Gel of a Serotonin Receptor Agonist, Buspirone Hydrochloride for the Brain Delivery via Intranasal Route.
Mathure D et al. Pharm Nanotechnol. 2018;6(1):69-78. doi: 10.2174/2211738506666180130105919
Chitosan Glutamate-Coated Niosomes: A Proposal for Nose-to-Brain Delivery.
Rinaldi F et al Pharmaceutics. 2018 Mar 22;10(2). pii: E38. doi: 10.3390/pharmaceutics10020038.
References cited in the sectiions “gold and silver nanoparticles” (77-88) are not strictly related to nose-to-brain delivery
Few examples of recent studies:
Intranasal delivery of targeted polyfunctional gold-iron oxide nanoparticles loaded with therapeutic microRNAs for combined theranostic multimodality imaging and presensitization of glioblastoma to temozolomide. Sukumar UK et al. Biomaterials. 2019 Oct;218:119342. doi: 10.1016/j.biomaterials.2019.119342.
In vivo comparisons of silver nanoparticle and silver ion transport after intranasal delivery in mice.
Falconer JL, Grainger DW. J Control Release. 2018 Jan 10;269:1-9. doi: 10.1016/j.jconrel.2017.10.018. Epub 2017 Oct 20.
Size-Dependent Deposition, Translocation, and Microglial Activation of Inhaled Silver Nanoparticles in the Rodent Nose and Brain. Patchin ES Environ Health Perspect. 2016 Dec;124(12):1870-1875.
Author Response
The paper seems to be successfully improved.
Some minor points have to be still addressed before publication.
>Authors added a section on niosomes but none of the cited references (55-59) are related to niosomes and nose-to-brain
Few examples of recent studies:
inPentasomes: An innovative nose-to-brain pentamidine delivery blunts MPTP parkinsonism in mice.
Rinaldi F et al J Control Release. 2019 Jan 28;294:17-26. doi: 10.1016/j.jconrel.2018. [61]
Formulation and Evaluation of Niosomal in situ Nasal Gel of a Serotonin Receptor Agonist, Buspirone Hydrochloride for the Brain Delivery via Intranasal Route.
Mathure D et al. Pharm Nanotechnol. 2018;6(1):69-78. doi: 10.2174/2211738506666180130105919 [63]
Chitosan Glutamate-Coated Niosomes: A Proposal for Nose-to-Brain Delivery.
Rinaldi F et al Pharmaceutics. 2018 Mar 22;10(2). pii: E38. doi: 10.3390/pharmaceutics10020038. [62]
Response: This section has been revised and reviewer’s suggested references have been cited. The number of the references has been given above in square brackets [].
>References cited in the sectiions “gold and silver nanoparticles” (77-88) are not strictly related to nose-to-brain delivery
Few examples of recent studies:
Intranasal delivery of targeted polyfunctional gold-iron oxide nanoparticles loaded with therapeutic microRNAs for combined theranostic multimodality imaging and presensitization of glioblastoma to temozolomide. Sukumar UK et al. Biomaterials. 2019 Oct;218:119342. doi: 10.1016/j.biomaterials.2019.119342 [85].
nanoparticle and silver ion transport after intranasal delivery in mice.
Falconer JL, Grainger DW. J Control Release. 2018 Jan 10;269:1-9. doi: 10.1016/j.jconrel.2017.10.018. Epub 2017 Oct 20. [94]
Size-Dependent Deposition, Translocation, and Microglial Activation of Inhaled Silver Nanoparticles in the Rodent Nose and Brain. Patchin ES Environ Health Perspect. 2016 Dec;124(12):1870-1875. [93]
Response: This section has been revised and reviewer’s suggested references have been cited. The number of the references has been given above in square brackets [].